# Acute Facial Nerve Palsy in Children: Gold Standard Management

**DOI:** 10.3390/children9020273

**Published:** 2022-02-17

**Authors:** Delphine Wohrer, Thomas Moulding, Luigi Titomanlio, Léa Lenglart

**Affiliations:** 1Department of Paediatric Emergency Care, APHP—Hôpital Robert Debré, 75019 Paris, France; luigi.titomanlio@aphp.fr (L.T.); lea.lenglart@gmail.com (L.L.); 2Department of Specialty and Integrated Medicine, Leeds Teaching Hospitals Trust, Leeds LS9 7TF, UK; thomas.moulding@nhs.net; 3Paediatric Migraine and Neurovascular Diseases Unit, APHP—Hôpital Robert Debré, 75019 Paris, France; 4Paris University, INSERM U1141, DHU Protect, 75019 Paris, France

**Keywords:** facial nerve palsy, idiopathic, children, Bell’s palsy

## Abstract

Facial nerve palsy (FNP) is a common illness in the paediatric emergency department. Missed or delayed diagnosis can have a serious impact on a patient’s quality of life. The aim of this article is to give a recent overview of this pathology in terms of the causes, diagnosis, red flag symptoms, complementary examinations, treatments and follow-up in the child population. In cases of acquired, acute onset and isolated FNP, Bell’s palsy can be assumed, and no further investigation is required. In any other scenario, complementary examinations are required. Treatment depends on the aetiology. Corticosteroids, in addition to antiviral medication, are recommended to treat Bell’s palsy whenever a viral infection is suspected. However, the lack of randomised control trials in the paediatric population does not allow us to comment on the effectiveness of these treatments. In all cases, treated or not, children have a very good recovery rate. This review emphasises the necessity of randomised control trials concerning this frequent neurological pathology in order to better treat these children.

## 1. Introduction

Facial nerve palsy is a frequent presentation in paediatric accident and emergency departments and, given its abrupt onset and often rapid progression, requires prompt diagnosis to limit the impact on morbidity and quality of life of patients [1].

To understand the pathophysiology of facial paralysis, it is essential to know the anatomy of the facial nerve. It is the seventh cranial nerve and is responsible for facial expression, lacrimation and salivation [2]. The course of the facial nerve is divided into two parts: the intracranial segment and the extracranial segment. During its intracranial segment, the facial nerve travels through the facial canal in the temporal bone from the pons to the internal acoustic meatus and exits the cranium via the stylomastoid foramen. The posterior auricular nerve, which provides the motor innervation around the ear, is the first extracranial branch of the facial nerve. The motor branches of the posterior digastric and stylohyoid muscles arise immediately. The motor root of the facial nerve continues into the parotid gland and terminates by splitting into five branches. The buccal branch, cervical branch, marginal branch, temporal branch and zygomatic branch innerves the muscles of facial expression. Thus, in addition to motor fibres, the facial nerve also provides autonomic and sensory innervation to the external auditory meatus, the tympanic membrane and the pinna of the ear, and taste to part of the tongue. 

A facial palsy may result either from a peripheral lesion (peripheral facial palsy or facial nerve palsy—FNP) of the facial nerve (VII) or from a central lesion involving the upper motor neuron, due to damage above the facial nucleus between the cortex and the pons (central facial palsy). Central facial palsy predominates on the lower part of the face and is more often associated with ipsilateral hemiparesis. Indeed, due to the bilateral innervation of the upper half of the face, the forehead muscles are spared in cases of upper motor neuron lesions, and only the contralateral side of the lower face is affected. On the other hand, a lesion in the lower motor neuron results in ipsilateral facial palsy involving both the upper and the lower face. In this study, we will focus on FNP.

Bell’s palsy is the idiopathic form of FNP, described in 1830 by Sir Charles Bell. It is an acquired, idiopathic facial palsy characterised by a palsy or weakness of facial muscles, usually on one side, with no obvious cause. This corresponds pathologically to symptoms including immobility of the brow, incomplete lid closure, drooping of the corner of the mouth, impaired closure of the lips, dry eye, hyperacusis, impaired taste, and pain around the ear [3]. It is an exclusion diagnosis, when no other cause can be identified. The challenge of FNP in a paediatric emergency department is being able to determine which patients require haematological investigations or imaging and which can be safely discharged without the need for further tests.

Thus, in this narrative review, we will present the paediatric causes of FNP and their epidemiology, delineate how to accurately diagnose patients and provide an overview of the current treatments commonly used in the management of FNP.

## 2. Methodology

We searched electronic databases such as PubMed, the US National Library of Medicine and the Cochrane Database of Systematic Reviews. The terms “facial nerve palsy” or “facial nerve paralysis” or “Bell’s palsy”, “children” or “paediatrics”, associated with “diagnosis”, “causes”, “management”, “treatment” and “follow-up”, were analysed for relevance. All English-language publications up to 1 June 2021 were screened. To screen for additional studies, we performed hand searches, using the bibliographies from pre-identified studies, review articles and guidelines. We selected 51 articles that were deemed relevant in providing an in-depth review of the gold standard management of facial palsy for paediatricians in 2022.

## 3. Epidemiology

An English-language study by Rowlands et al. (2002) reports different annual incidences of FNP in the general population: 13–28 per 100,000 based on population studies, utilising a detailed review of medical records; 16–24 per 100,000 based on the results from hospital-based studies; and 30 per 100,000 based on statistics from general practice [4]. This is supported by other studies which estimate that 15–40 per 100,000 adults are affected annually by FNP [5,6].

The incidence of FNP in children younger than 10 years of age is 2.7 per 100,000 annually, while in children 10 to 20 years of age, it is estimated to be 10.1 per 100,000 [4]. Other studies report an incidence rate between 5 and 21 per 100,000 children annually [7,8]. The mean age of onset was 6.6 years in a study of 29 children by Chen et al., compared with 9.2 years in a study of 106 children younger than 15 year by Jenke et al., with no significant difference between the sexes [8,9,10].

## 4. Causes of Facial Nerve Palsy

In the children population, the most common causes of FNP are Bell’s palsy (60 to 80% of the cases) [9,11,12], infectious diseases, malignancies, trauma and congenital abnormalities [13,14,15]. Therefore, these are the causes to investigate in the history and physical examination.

### 4.1. Bell’s Palsy

The prevalence rate of Bell’s palsy is debated; a retrospective study by Yilmaz et al. included 81 children diagnosed with peripheral facial palsy (aged 1–16 years) in a Turkish hospital between 2011 and 2013 and reported an idiopathic cause rate of 80.2% [11]. This is coherent with a retrospective study by Shih et al., who observed an idiopathic cause in 78.6% of cases (56 children, aged from birth to 15 years), from 1996 to 2002 in Taiwan [12]. Drack et al. reported an idiopathic cause in 60.7% of children diagnosed with peripheral facial palsy (84 patients aged 10 months to 16 years) retrospectively between 1998 and 2007 in Switzerland [9]. The reactivation of herpes simplex virus-1 is presumed to support the pathogenesis of Bell’s palsy [5,13]. Symptoms are likely caused by facial nerve oedema, with the autoimmune system proposed to be involved in causing local myelin damage and an inflammatory process of the facial nerve, leading to its compression [11,16,17]. To our knowledge, only Khine et al. studied the association between Bell’s palsy and HSV-1 (HSV-1 in children). In total, 33 of 42 affected patients had a positive HSV-1 enzyme-linked immunosorbent assay (ELISA) compared with 16 of 41 controls (*p* = 0.0003). This underlines the potential association between HSV-1 and Bell’s palsy in children [18].

### 4.2. Infectious Causes

Among the infectious causes, the first to investigate is acute otitis media. Its incidence rate varies from 4% to 37% of all peripheral nerve palsies [11,19,20]. FNP can also complicate an acute mastoiditis or chronic otitis media. However, the prevalence of FNP as a sequela of acute otitis media has decreased [15] as a consequence of vaccination and early antibiotic treatment.

The reactivation of herpes varicella-zoster virus, responsible for Ramsay Hunt syndrome, causes a facial palsy associated with vesicular lesions in the external auditory canal and the concha. The incidence of this syndrome under 10 years of age is reported to be 2.7 per 100,000 [21,22,23].

In areas where *Borrelia burgdorferi* infection is endemic, Lyme disease is the most common cause of acute facial paralysis in children [21,24]. Therefore, it should not be neglected in regions such as Central and Eastern Europe.

### 4.3. Other Causes

Direct trauma to the facial nerve may also cause palsy, for instance, a blunt force fracturing the temporal bones [25,26].

Lastly, paediatric facial nerve paralysis has also been described and associated with more severe pathologies such as leukaemia, posterior fossa tumours and parotid gland tumours, which are very important not to miss [21,27].

Rarer causes of FNP found in the scientific literature are outlined in Table 1.

## 5. Diagnosis

### 5.1. History and Physical Examination

In patients presenting with unilateral facial paralysis, the clinician should ascertain a thorough history, followed by a systematic examination to identify possible causes and exclude those that are inconsistent with their findings [21,28]. A complete neurological examination, otoscopy and blood pressure measurement are mandatory [29]. Any comorbidity affecting the child should be investigated and clinicians should also assess the onset and duration of symptoms, since a gradual onset may suggest a neoplastic cause such as a mass lesion. Clinicians should also enquire about a history of tick bites or outdoor pursuits in tick endemic regions which might raise the clinical suspicion of Lyme disease. Furthermore, a history of trauma warrants investigation for other stigmata of temporal bone fractures during the physical examination. Physical examination should initially focus on any other neurological symptoms that may suggest a malignancy, as well as an examination of the neck to look for lymph nodes or a mass. Following this, an ENT examination should be conducted, paying particular attention to the inspection of the external auditory canal, eardrum and the mastoid region; causes such as acute otitis media, cholesteatoma, mastoiditis or Ramsay Hunt syndrome (if a vesicular rash is present) may be identified at this stage. Clinicians should look for the characteristic signs of Lyme disease, such as erythema migrans or arthritis. Finally, blood pressure measurement is required because it has been noted that high blood pressure in children can be associated with recurrent facial palsy [23].

In conclusion, the clinical examination should focus on the various signs that may point to underlying causes of FNP that require specific treatment; these signs are summarised in Table 2.

### 5.2. Grading System

The severity of FNPs must be estimated using a grading system [21] as it aids in the assessment of recovery or progression in medical follow-up clinics. The House-Brackmann facial nerve grading system is one of the most commonly used tools for clinical assessment. The scale is based on functional impairment, ranging from I (normal) to VI (no movement), summarised in Table 3 [30,31].

### 5.3. Complementary Examinations

#### 5.3.1. Laboratory Testing

The recommendation of the American Academy of Otolaryngology—Head and Neck Surgery Foundation (AAO-HNSF) is that routine laboratory investigation is not required in patients newly presenting with Bell’s palsy [28].

Nevertheless, laboratory tests are mandatory in certain situations:A full blood count and a blood film should be performed in cases of suspected leukaemia based on examination [29];Serological testing for Lyme disease is indicated in cases suggestive of possible tick exposure, in endemic areas, or for patients with erythema migrans [28];A lumbar puncture should be conducted when suspecting meningitis, Lyme disease or Guillain-Barré syndrome [21].

#### 5.3.2. Diagnostic Imaging

Medical practitioners who strongly suspect Bell’s palsy as the cause for facial paralysis do not need to request radiological imaging for confirmation of the diagnosis [28]. Although magnetic resonance imaging (MRI) often shows facial nerve enhancement, it does not influence the medical management and therefore it is not recommended [32,33].

Nevertheless, imaging is still required for patients with recurrent paralysis, or when the symptoms are complex and do not completely correspond to Bell’s palsy, or when there has been no improvement after 3 weeks of evolution. Imaging is also essential in cases where additional neurological abnormalities are observed, or in suspected malignancy [21,34].

#### 5.3.3. Electrodiagnostic Testing

It is not recommended to perform electrodiagnostic testing in patients with Bell’s palsy as it cannot successfully predict recovery and therefore has no clinical utility. Moreover, studies show that a reduction in the amplitude of motor responses is only seen in patients presenting with significant muscle weakness, giving the test a low sensitivity [28,35].

## 6. Treatments

The treatment of facial palsy in children depends on its aetiology, as well as its grade of severity. We will focus on the treatment of idiopathic palsy in the paediatric population.

### 6.1. Corticosteroids

There is a lack of data concerning the use of corticosteroid to treat FNP in children. In fact, two paediatric systematic reviews analysing the use of steroids to treat Bell’s palsy (from 2012 and 2001) found no placebo-controlled trials. Moreover, both reviews concluded that there is insufficient evidence to make a treatment recommendation [36,37].

Nevertheless, it seems that steroids may hasten the time to recovery. Unüvar et al. conducted a randomised control trial of 42 children, who received methylprednisolone (1 mg/kg/day) or no placebo. Although all children recovered fully within 12 months regardless of treatment, more children in the methylprednisolone group had recovered at 4 and 6 months (86% vs. 72% at 4 months and 100% vs. 89% at 6 months) [38]. Moreover, a large-scale recent paediatric study led by Hanci et al. (113 patients; 2019) established a statistically significant difference between children who received corticosteroids and their untreated counterparts when assessing mean time to recovery (*p* ≤ 0.01); overall, more than one-third of children who were treated with steroids made a full recovery within one month, whereas only one-eighth in in the control population did [39].

As a result of this paucity of data, two randomised, double-blind, placebo-controlled trials have recently commenced, which aim to determine the efficiency of corticosteroid treatment in children with Bell’s palsy [40,41]. The first one (FACE: facial nerve palsy and cortisone evaluation) will include 500 children in Sweden, between 1 and 17 years old. No interim analysis will be made, and the results are expected in 2024. The second one (BellPIC: Bell’s Palsy in Children), in Australia and New Zealand, will include 540 children aged between 6 months and 18 years. Patients will receive either prednisolone or a placebo for 10 days.

By comparison, in the adult population, short-term corticosteroid treatment, initiated within the first 72 h, is widely recommended to treat idiopathic FNP [28]. Thus, on children, if corticosteroids are used, it is recommended to begin preferably within 3 days (up to 7 days) from onset of symptoms [36]. The recommendation is to start oral prednisolone at a dose of 1 to 2 mg/kg/day (maximum dose 60 mg) for 5 to 7 days, and then stop [10,36,42]. However, regarding administration of corticosteroids and their dosing regimens, caution should be used in patients with diabetes mellitus, hypertension, renal or hepatic dysfunction or an underlying immunocompromised state [43]. The two randomised trials currently in progress will likely help to determine the efficacy of corticosteroid treatment in children.

### 6.2. Antivirals

In the adult population, a review and meta-analysis from 2009 suggests that the addition of an antiviral therapy does not provide any benefit (OR, 1.03 [95% CI, 0.74–1.42]; *p* = 0.88) at 3, 4, 6 and 9 months of assessment. This study reviewed 738 patients with Bell’s palsy: 372 in the corticosteroid group and 366 in the combined therapy group, in which the regimen included a corticosteroid plus an antiviral (aciclovir or valaciclovir) [44].

Even in the adult population, the recommendations are unclear on the use of antivirals: the AAO-HNSF (2013) recommends, alongside steroids, the ancillary use of antivirals for all cases who present within three days of onset, whereas the Canadian guidelines (2014) recommend corticosteroid and antiviral therapy only for patients with severe paralysis [28,45]. The latest Cochrane review from 2019 (14 trials, including 2488 participants) concludes that a combination of antivirals and corticosteroids probably reduces the late sequelae of Bell’s palsy compared with corticosteroids alone [46].

Interestingly, in the paediatric population, patients who had positive serological findings (Epstein-Barr virus, varicella-zoster virus, mycoplasma pneumoniae, herpes simplex virus) had a prolonged recovery time in a study from 2005, based on 29 children. Ten children with a positive serological finding recovered in 3.36 ± 2.12 months and 22 children with negative serological finding recovered in 1.69 ± 1.12 months (*p* ≤ 0.007) [10].

Moreover, in a recent retrospective paediatric study (from 2011 to 2015), in addition to corticosteroids clinicians added aciclovir or valaciclovir (30 mg/kg/d) when a viral infection was suspected. A complete recovery of 98% was recorded in 124 children [1]. These findings suggest that treatment should be adapted to each patient; if no infectious cause is suspected, antiviral treatment should not be used.

### 6.3. Eye Care

Because of facial palsy, the eyelid on the affected side may not close completely and so the eye is at risk of irritation and corneal ulceration [5]. The eye is both unprotected and dry because of improper lid closure and decreased tear production.

The use of eye-protective measures is therefore strongly recommended [28,45]. These include the administration of artificial tears numerous times (ideally hourly) throughout the day, ophthalmic lubricating ointment during sleep, an eyelid patch at night and sun protection [14,17,47]. Adequate and appropriate patient education is also required.

An ophthalmological opinion is desirable in case of a long evolution or corneal damage [3].

### 6.4. Physiotherapy

The Cochrane review from 2011 assessed 12 studies (872 participants) and found no evidence to support any benefit or deterioration following any physical therapy for idiopathic facial paralysis [48]; therefore, it is not currently recommended for children.

## 7. Prognosis

The spontaneous recovery rate in children with Bell’s palsy is excellent (up to 90% at 6 months and nearly 100% at 1 year) [14].

There are numerous studies examining the outcome of this disease in children. For example, in 2005, Chen et al. described a recovery rate of 68.8% at 3 weeks and 96.9% at 7 months in a population of 29 children in Hong Kong in a retrospective study; the authors did not find a relationship between the evolution and the treatment received by the patients [10]. In 2009, Shih et al. described a recovery rate of 97.7% among 44 children in Taiwan [12]. Furthermore, in 2011, Jenke et al. described a recovery rate of 97.6% within 12 weeks among 106 children in Germany, even though only 3.7% of them received corticosteroids [8]. More recently, Drack et al. showed a recovery rate of 89.3% among 84 children in Switzerland and only 4 had received corticosteroids [9].

Overall, the spontaneous recovery rate of Bell’s palsy in paediatric populations can be as high as 97%. As a result, it remains challenging to provide the conclusive evidence needed to recommend steroids in children. By contrast, incomplete recovery is seen in approximately 20% of adults, which, following risk-benefit analysis, suggests a clear advantage in favour of steroid administration [20,24].

## 8. Follow-Up

As previously stated, Bell’s palsy has an excellent prognosis in the paediatric population; most children recover fully by 6 weeks, though some can take up to a year [7,49].

Possible chronic complications of FNP are persisting weakness, synkinesis, spasms, autonomic dysfunction (decreased tearing, crocodile tears), corneal abrasion and psychological consequences [2,24]. Recurrent FNP is uncommon in children; it has been reported in only 6% of cases [11].

Although the functional recovery in general is excellent, a slight face asymmetry may persist on close inspection in up to 50% patients, according to Biebl et al., who conducted a study of 56 children. Although it is a cosmetic issue, social impact can be significant. A child with FNP may feel more hesitant to go out in public, avoid participating in play with other children, or shy away from interacting. Therefore, this condition results in functional and aesthetic problems for the child and their family and it is important to provide psychological support to the child and their parents [2,50].

A follow-up consultation should be performed by a general practitioner or a paediatrician to assess for resolution of symptoms. In all cases, clinicians should reassess, or refer to a facial nerve specialist, Bell’s palsy patients with:ocular symptomsnew or worsening neurological symptomsincomplete facial recovery after 3 months of evolution [28].

Moreover, if there is any doubt, it may be reasonable to consult an otolaryngologist to confirm the diagnosis and to exclude differential diagnoses [45].

## 9. Future Directions

This study emphasises the necessity of randomised control trials in order to assess the efficacy of corticosteroids and other therapeutics to treat Bell’s palsy in children, as well as the need for further research on prognostic factors of poor recovery, which may help determine indications for treatment.

In conclusion, FNP is a fairly common disease in the paediatric emergency department and its annual incidence rate is approximately 10/100,000. Thus, paediatricians and general physicians need to be cognizant of its diagnosis and management. A diagnosis of Bell’s palsy can be reliably established in cases of acquired, acute onset and isolated FNP without the need for further laboratory or radiological investigation. As of now, its management includes corticosteroids with or without antiviral therapy in a population with very good recovery rates. 

## Figures and Tables

**Table 1 children-09-00273-t001:** Causes of facial nerve palsy.

Causes	Examples
Idiopathic VII nerve palsy (Bell’s palsy)	Isolated—acute onset—unilateral—detailed history and examination are normal
Infectious	Otitis media (4 to 37%) [11,19,20] Mastoiditis or cholesteatoma Herpes zoster (Ramsay Hunt syndrome) Epstein-Barr virus Lyme disease Tuberculosis Cytomegalovirus Adenovirus Rubella Mumps Human immunodeficiency virus (HIV) Haemophilus influenza Mycoplasma pneumoniae Syphilis Leprosy Cat scratch fever
Neoplasia/malignancy	Posterior fossa tumours Parotid gland tumours Leukaemia Lymphoma Cholesteatoma
Trauma/nerve compression	Perinatal trauma Temporal bone fracture Raised intracranial pressure Otic barotrauma Iatrogenic trauma (surgical procedures) Cleidocranial dysostosis Histiocytosis X
Congenital/genetic	Arnold-Chiari syndrome Absence of depressor anguli oris muscle (cardiofacial syndrome) Inner ear or facial nerve malformation Moebius syndrome Syringobulbia Facioscapulohumeral muscular dystrophy Myotonic dystrophy Myasthenic syndromes
Neurological	Guillain Barré syndrome Multiple sclerosis
Inflammatory	Henoch-Schönlein purpura Kawasaki disease Sarcoidosis
Metabolic conditions	Diabetes mellitus Hyperparathyroidism Hypothyroidism Acute porphyria
Other	Hypertension Autoimmune issues (e.g., lupus) Pregnancy Haemophilia Melkersson-Rosenthal syndrome

**Table 2 children-09-00273-t002:** Characteristic signs of specific causes of facial palsy.

Signs	Possible Causes
History of trauma	Search for stigmata of temporal bone fractures: haemotympanum, traumatic perforation, Battle’s sign
Gradual onset	Malignancy
Tick bites or possible exposure	Lyme disease
Erythema migrans	Lyme disease
Arthritis	Lyme disease
Bilateral involvement	Lyme disease, polyneuropathy
Forehead sparing	Central nervous system cause
Abnormal otoscopy	Acute otitis media, cholesteatoma
Fever	Infectious cause (acute otitis media)
Vesicular rash or blistering of the face or ear canal	Ramsay Hunt syndrome, herpes zoster virus infection
Ear pain	Herpes zoster virus infection
Rest of the neurological examination abnormal	Malignancy
Examination of the neck: lymph nodes or mass	Malignancy
High blood pressure	Malignant hypertension

**Table 3 children-09-00273-t003:** House-Brackmann facial nerve grading system.

Stage	Description
I	No paresis: normal function
II	Mild paresis: no deformity at rest
III	Moderate paresis: obvious difference from the other side, no deformity at rest, synkinesis, total closure of eyelids at maximum effort
IV	Moderately severe paresis: disfiguring asymmetry, synkinesis, eye closure incomplete at maximum effort
V	Severe paresis: asymmetry at rest (ptosis of the labial commissure, effacement of the nasolabial fold), some visible residual movements
VI	Complete paralysis: atony at rest, no active movement

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
