# Peer review of "Acute Facial Nerve Palsy in Children: Gold Standard Management"

_children, 2022, doi:10.3390/children9020273_

Round 1
Reviewer 1 Report
The work is globally very well written and well divided into different paragraphs. Due to its clarity, the article is very useful for pediatricians or students approaching this topic: it could represent a guide with many appropriate citations from the main updated guidelines.
The only weak part is paragraph number 3: causes of facial nerve palsy. It includes a list of the most relevant causes of facial nerve palsy but the order in which there are listed is unclear. I think that the authors should list them based on the evidence in literature as in the table 1, listing the most frequent first based on the incidence rate. For example, HSV-1 is cited first, while otitis media seems to represent the most frequent cause.
Reviewer 2 Report
Dear authors:
The manuscript is interesting and the topic is important, but the parts of manuscript are not elegibled. It is very difficult to understand how was it designed? What is the methodology? what is the manuscript type: review, narrative review....how the data was obtained? what are the articles? Why these points?
The conclusion is poor
I suggest you re-written the article and clarify the points in a correct way, where the reader cand understand your work.
Round 2
Reviewer 2 Report
Felicitaciones por el trabajo. El manuscrito ha sido mejorado de acuerdo a las sugerencias